

# The response of sulfur dioxygenase to sulfide in the body wall of *Urechis unincinctus*

Litao Zhang[1,2] and Zhifeng Zhang[2]

[1] School of Biological Science, Jining Medical University, Rizhao, China
[2] Key Laboratory of Marine Genetics and Breeding, Ministry of Education, Ocean University of China, Qingdao, China

## ABSTRACT

**Background**. In some sedimentary environments, such as coastal intertidal and subtidal mudflats, sulfide levels can reach millimolar concentrations (2–5 mM) and can be toxic to marine species. Interestingly, some organisms have evolved biochemical strategies to overcome and tolerate high sulfide conditions, such as the echiuran worm, *Urechis unincinctus*. Mitochondrial sulfide oxidation is important for detoxification, in which sulfur dioxygenase (SDO) plays an indispensable role. Meanwhile, the body wall of the surface of the worm is in direct contact with sulfide. In our study, we chose the body wall to explore the SDO response to sulfide.

**Methods**. Two sulfide treatment groups (50 µM and 150 µM) and a control group (natural seawater) were used. The worms, *U. unincinctus,* were collected from the intertidal flat of Yantai, China, and temporarily reared in aerated seawater for three days without feeding. Finally, sixty worms with similar length and mass were evenly assigned to the three groups. The worms were sampled at 0, 6, 24, 48 and 72 h after initiation of sulfide exposure. The body walls were excised, frozen in liquid nitrogen and stored at −80 °C for RNA and protein extraction. Real-time quantitative RT-PCR, enzyme-linked immunosorbent assay and specific activity detection were used to explore the SDO response to sulfide in the body wall.

**Results**. The body wall of *U. unincinctus* consists of a rugal epidermis, connective tissue, outer circular muscle and middle longitudinal muscle. SDO protein is mainly located in the epidermis. When exposed to 50 µM sulfide, SDO mRNA and protein contents almost remained stable, but SDO activity increased significantly after 6 h ($P < 0.05$). However, in the 150 µM sulfide treatment group, SDO mRNA and protein contents and activity all increased with sulfide exposure time; significant increases all began to occur at 48 h ($P < 0.05$).

**Discussion**. All the results indicated that SDO activity can be enhanced by sulfide in two regulation mechanisms: allosteric regulation, for low concentrations, and transcription regulation, which is activated with an increase in sulfide concentration.

Corresponding authors
Litao Zhang,
zhanglitao@mail.jnmc.edu.cn
Zhifeng Zhang, zzfp107@ouc.edu.cn

## INTRODUCTION

Although hydrogen sulfide ($H_2S$) at low concentrations can act as a biological signaling molecule in many physiological processes, including regulation of vascular tone, cellular stress response, apoptosis, and inflammation (*Li, Rose & Moore, 2011*; *Módis, Wolanska & Vozdek, 2013*; *Kabil & Banerjee, 2014*), $H_2S$ is inherently toxic at high concentrations by reducing complex IV activity, lowering the electrochemical potential across the inner mitochondrial membrane, reducing ATP generation and inducing apoptosis (*Beauchamp et al., 1984*; *Jiang et al., 2016*). In brief, $H_2S$ is a mitochondrial inhibitor. $H_2S$ in solutions usually exists in forms of $H_2S$, $HS^-$ and $S^{2-}$, summarized as sulfide. There are many environments rich in sulfide in nature, where abundant biological species exists (*Hand & Somero, 1983*). In some sedimentary environments, such as coastal intertidal and subtidal mudflats, sulfide levels can reach millimolar concentrations (*Arp, Hansen & Julian, 1992*), in which animals living there have a variety of adaptations to avoid the toxicity of sulfide. Sulfide detoxification, especially mitochondrial sulfide detoxification, is one of the most important strategies to detoxify sulfide (*Grieshaber & Völkel, 1998*). The enzyme systems including sulfide:quinine oxidoreductase (SQR), sulfur dioxygenase (SDO) and sulfur transferase (ST) take part in sulfide oxidation and convert sulfide to harmless thiosulfate, in which SDO plays an indispensable role in catalyzing persulfide oxidized to sulfite (*Hildebrandt & Grieshaber, 2008*; *Jackson, Melideo & Jorns, 2012*).

The *ETHE1* (*ethylmalonic encephalopathy 1*) gene in humans was identified as the *SDO* gene, and its dysfunction can lead to a fatal autosomal recessive mitochondrial disease: ethylmalonic encephalopathy (*Tiranti et al., 2009*). Subsequently, the biochemical characterization of SDO and its kinetic properties were determined for humans and *Arabidopsis thaliana* (*Kabil & Banerjee, 2012*; *Holdorf et al., 2012*). In rice, the ETHE1 promoter was cloned, and its activity was induced by various abiotic stresses (*Kaur et al., 2014*). In *Acidithiobacillus caldus*, two SDOs were identified; one was essential for the survival of *A. caldus* and involved energy supply, while the other might function in sulfur oxidation (*Wu et al., 2017*). However, few studies were conducted on the SDO response to sulfide.

The echiuran worm *Urechis unicinctus*, which inhabits a U-shaped burrow in coastal intertidal and subtidal mudflats, can tolerate sulfide and detoxify it through oxidation in mitochondria (*Ma et al., 2012a*; *Ma et al., 2012b*). The function and expression characteristics analysis of the SDO gene in *U. unicinctus* has been conducted before (*Zhang et al., 2013*; *Zhang et al., 2016*). In *Urechis caupo*, the body was recognized as an exchange surface and as a permeation barrier, but ultrastructural studies found that the body wall does not present a significant structural barrier to permeation, so oxidation of sulfide in the body wall might play important roles (*Menon & Arp, 1993*). However, the responses of the enzyme involved in sulfide oxidation to sulfide in the body wall is unknown. Thus, in this study, the body wall was chosen to explore the SDO response to sulfide through RNA, protein and enzyme activity levels. Our aim is to answer the question 'how does SDO gene expression, protein concentration, and activity respond to sulfide concentrations in
**Table 1  Sequences of designed primers used in this study.**

| Primer | Sequence | Product length |
|---|---|---|
| SDO-F | 5′-ACAGGGATGTTCGTATCGTCAA-3′ | 165 bp |
| SDO-R | 5′-ATTCGGCAATCACACTCTTACG-3′ | |
| β-actin-F | 5′-CACACTGTCCCCATCTACGAGG-3′ | 153 bp |
| β-actin-R | 5′-GTCACGGACGATTACACGCTC-3′ | |

the body wall' and to reveal the function of the body wall in sulfide detoxification at the molecular level.

# MATERIALS AND METHODS

## Sulfide treatment and sampling

*U. unicinctus* were collected from the intertidal flat of Yantai, China. Upon arrival at the lab, the worms were temporarily reared in aerated seawater (18 °C, pH 8.0, and salinity 30‰) for three days without feeding. Then, sixty worms with similar length and mass were evenly assigned to six tanks containing 30 L of seawater and sealed with cling film. Three groups, including a control group without sulfide and two sulfide treatment groups (50 μM and 150 μM) were used in this study. During the experiment, the sulfide concentrations were maintained by adding a sulfide stock solution (10 mM $Na_2S$, pH 8.0) every 2 h as necessary, based on the determined sulfide concentration by the methylene blue method. The times for sampling were set at 0, 6, 24, 48 and 72 h after initiation of sulfide exposure. The body walls were excised, frozen in liquid nitrogen and stored at −80 °C for RNA and protein extraction.

## RNA isolation and qRT-PCR

Total RNA from the body wall of *U. unicinctus* was extracted by the TRIzol reagent (Invitrogen, Carlsbad, CA, USA) according to the manufacturer's instructions. The quality of the RNA samples was assessed by a NanoDrop microvolume spectrophotometer (Thermo Scientific, Waltham, MA, USA) and by electrophoresis using a 1.2% agarose gel. The cDNA templates were obtained using a PrimeScript RT reagent Kit with gDNA Eraser (Takara, Otsu, Japan). The expression pattern of SDO was determined by qRT-PCR and normalized with the reference gene β-actin (GenBank accession number GU592178.1). All the primers used in the study are listed in Table 1. qRT-PCR was performed in a 7500 Real-Time PCR System (ABI, CA, USA) with a 20 μL reaction volume containing 2 μL of template cDNA, 0.8 μL of each primer (10 μM), 0.4 μL of 50× ROX Reference Dye II, 10 μL of 2× SYBR Premix Ex Taq (Takara, Otsu, Japan) and 6 μl of PCR-grade water. Each reaction was performed in quadruplicates. The relative expression levels of SDO were analyzed according to the $2^{-\Delta\Delta CT}$ method.

## Antibody preparation and ELISA

The SDO recombinant protein of *U. unicinctus* has previously been successfully obtained by the PET Express System (*Zhang et al., 2013*). SDO protein was expressed in the form of inclusion bodies and therefore dissolved in 8 M urea and purified by Ni-NTA affinity
chromatography (Novagen, Darmstadt, Germany). The purified SDO protein was used to produce rabbit-anti-*U. unicinctus* polyclonal antibody by Sangon Biotechnology (Shanghai, China). The specificity of the SDO antibody was validated by western blot and deemed appropriate for ELISA to determine SDO protein contents. The protein from different tissues were extracted using a Tissue Protein Extraction Kit (CWBIO, China). Tissue (0.1 g) was placed into 1 ml of tissue protein extraction reagent (50 mM Tris-Cl, 150 mM NaCl, 1% NP 40 and 0.5% sodium deoxycholate) with 10 μL of the protease inhibitor cocktail, and then the sample was homogenized and placed on ice for 20 min. After centrifugation at 12,000 rpm for 30 min, tissue protein was obtained, and its concentration was determined by the Coomassie brilliant blue method. The indirect competitive ELISA method was established previously, and the SDO contents in the body wall were calculated as described previously (*Zhang et al., 2016*). Each reaction was performed in quintuplicate.

## Immunohistochemistry

To assess the SDO location in the body wall, paraffin sections (7 μm) were first cut using a Histostart 820 Rotary microtome (Reichert, Depew, NY, USA). Then, the sections were dewaxed in xylene and gradient alcohol. After that, antigen retrieval for the sections was conducted with the procedure in 3% $H_2O_2$ at room temperature for 15 min and in EDTA (0.05 M pH 8.0) at 85 °C for 1 h. Nonspecific protein binding was blocked with 3% bovine serum albumin (BSA) for 30 min. The sections were incubated in rabbit anti-SDO antibody (diluted 1:500) or preimmune serum, as the negative control, for 1 h. Then, incubation was followed by washing the tissue section followed by a 30 min incubation with HRP-labeled goat anti-rabbit IgG (diluted 1:1,000) (Sangon). The staining was completed by the use of an HRP Color Development Kit (Solarbio, Shanghai, China). Next, sections were counterstained with hematoxylin, dehydrated and mounted in Pertex. A Nikon E80i microscope (Nikon, Tokyo, Japan) was used to observe and photograph the sections.

## Analysis of SDO specific activity

Total SDO specific activity was determined by the improved *Kabil & Banerjee (2012)*. The reaction mixture (2 ml) contained 1 ml of potassium phosphate buffer (0.2 M, pH 7.4), 200 μL of 10 mM reduced glutathione, 30 μL of a saturated acetonic sulfur solution and 750 μL of sterile water. Twenty microliters of the total organ protein were added to initiate the reaction, and the Oxytherm oxygen measurement system (Hansatech, Pentney, UK) was used to record the rate of $O_2$ consumption at 25 °C. The $O_2$ consumption rate per mg total organ protein (1 μmol of $O_2$ min mg total protein$^{-1}$) was used to represent the SDO specific activity from the total protein (U mg total protein).

## Statistical analysis

All parametric data are expressed as the mean ± standard error (SE). Statistical comparisons among means were tested by one-way ANOVA (analysis of variance), and a value of $P < 0.05$ was considered significant (computed by SPSS version 18.0 for Windows).

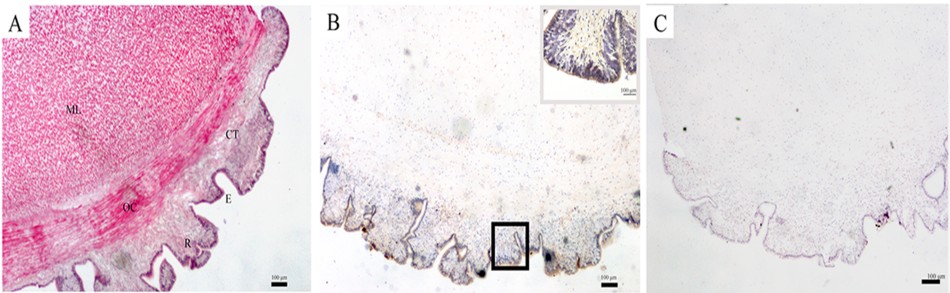

**Figure 1** **SDO protein distribution in the body wall of *U. unicinctus*.** (A) Hematoxylin-eosin (H&E) stain; (B) Positive results of immunohistochemistry, insets, magnification of the box in the same picture; (C) Negative control. The scale bars in the insets of images are 100 nm. Abbreviation: CT, connective tissue; E, epithelial tissue; ML, middle longitudinal muscle; OC, outer circular muscle; R, rugae.

## RESULTS

### SDO distribution in the body wall

The structure of the body wall, as observed by a light microscope, is shown in Fig. 1A. The surface of the body wall is epidermis with abundant rugae; subjacent to the epidermis is a thin layer of connective tissue, and below it are two distinct muscle layers: an outer circular muscle and a middle longitudinal muscle. The immunohistochemistry results showed that SDO protein was mainly expressed in epithelial tissue (Fig. 1B). Moreover, the negative control had no positive signals (Fig. 1C), which provides assurance that the results are accurate.

### Changes in *SDO* mRNA levels after sulfide exposure

The temporal expression levels of SDO mRNA in the different treatment groups were examined at 0, 6, 24, 48 and 72 h by qRT-PCR (Fig. 2). In the 50 µM sulfide treatment group (Fig. 2), no significant changes were observed during different sulfide exposure times. However, in the 150 µM sulfide treatment group (Fig. 2), the expression level of SDO mRNA was upregulated at 48 h (1.885-fold, $P < 0.05$) and 72 h (2.183-fold, $P < 0.05$). Furthermore, there was no obvious change in the control group during the whole experimental process.

### Expression pattern of SDO protein after sulfide exposure

The SDO protein contents in different treatment groups were determined by ELISA at 0, 6, 24, 48 and 72 h (Fig. 3). The SDO contents increased with the time of sulfide exposure, but no significant difference occurred during the experiment in the 50 µM sulfide treatment group (Fig. 3). In the 150 µM sulfide treatment group (Fig. 3), a distinct time-dependent elevation of the SDO protein level was observed compared with the SDO contents at 0 h ($1.103 \pm 0.085$ ng µg total protein$^{-1}$), and a significant increase in the SDO level ($P < 0.05$) was observed at 48 h ($1.723 \pm 0.076$ ng µg total protein$^{-1}$) and at 72 h ($2.165 \pm 0.079$ ng µg total protein$^{-1}$) after sulfide exposure. Moreover, no obvious change in the control group was observed during the whole experiment.

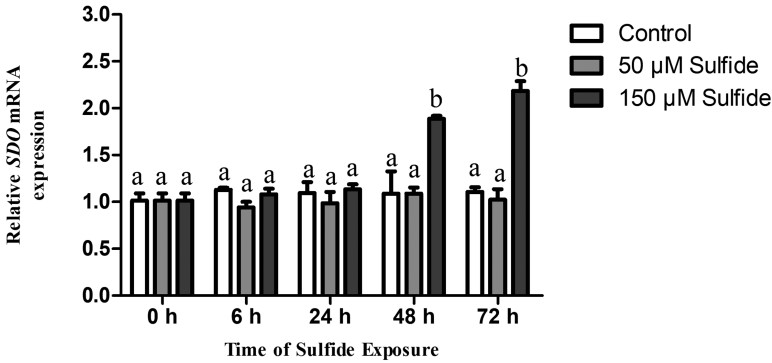

**Figure 2  Relative expression levels of SDO in different sulfide treatment groups.** The data are mean ± S.E. for each group ($n = 4$). Differing letters indicate significant differences among different groups ($P < 0.05$); identical letters indicate no significant difference ($P > 0.05$).

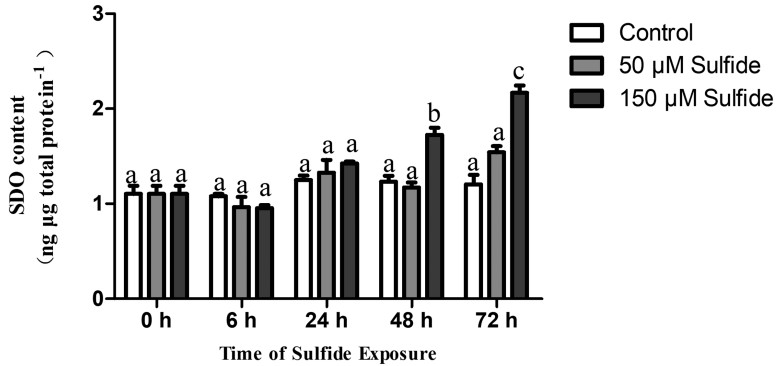

**Figure 3  SDO protein contents expression pattern in different sulfide treatment groups.** The data are mean ± S.E. for each group ($n = 4$). Differing letters indicate significant differences among different groups ($P < 0.05$); identical letters indicate no significant difference ($P > 0.05$).

### SDO activity responses to sulfide

The T-SDO SA (total SDO specific activity from total protein) was detected using a Clark oxygen electrode at 0, 6, 24, 48 and 72 h in different groups (Fig. 4). No significant differences in T-SDO SA ($p > 0.05$) were observed at the various detected times for the control group (Fig. 4). The T-SDO SA was elevated gradually with the delay in sulfide exposure time in both the 50 μM and 150 μM sulfide treatment groups; in 50 μM sulfide treatment group, the significant increase in T-SDO SA ($p < 0.05$) occurred at 6 h, reaching $0.130 \pm 0.012$ U mg total protein$^{-1}$, and then became stable, while the T-SDO SA was elevated significantly ($p < 0.05$) at 48 h ($0.159 \pm 0.002$ U mg total protein$^{-1}$) with the presence of a significant increase at 72 h (Fig. 4).

### DISCUSSION

The body wall, which is located at the surface of *U. unicinctus*, is in direct contact with sulfide in the environment. Therefore, it has its own special histological adaption for the

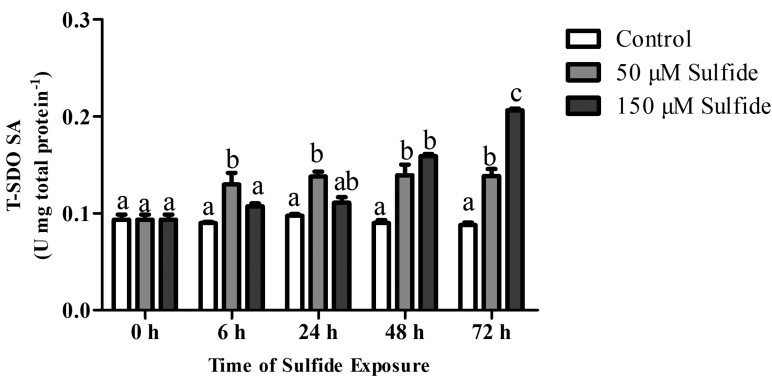

**Figure 4** **SDO activity response to sulfide in different groups.** The data are mean ± S.E. for each group ($n = 6$). Differing letters indicate significant differences among different groups ($P < 0.05$); identical letters indicate no significant difference ($P > 0.05$).

protection of internal organs. HE staining of histological sections (Fig. 1A) revealed that the epidermis is in the form of rugae. Therefore, the surface area for oxygen uptake increases to ensure the oxygen supply in a sulfide-rich environment. The muscle in the body wall is thick and sinewy (Fig. 1A), which is helpful for the peristaltic movements of the body wall to obtain oxygen and feeding currents in the burrow. A similar structure has also been illustrated in *U. caupo* (*Menon & Arp, 1993*). It has been reported that there are no tight junctions in invertebrate epidermis except for tunicates (*Lillywhite & Maderson, 1988*). In *U. unicinctus*, no tight junctions were also observed. Thus, it was concluded that sulfide can permeate the body wall of *U. unicinctus*, and therefore, the sulfide should be detoxified by the organism itself. Sulfide oxidation in mitochondria is the most important strategy for sulfide detoxification.

In mitochondrial sulfide oxidation, SDO can oxidize persulfide to sulfite in the presence of oxygen, which is indispensable. The persulfide is produced by SQR with the substrates -sulfide and GSH (*Hildebrandt & Grieshaber, 2008*) or by ST with the substrates thiosulfate and GSH (*Jackson, Melideo & Jorns, 2012*). In our study, we found that SDO protein was mainly located in the epidermis of the body wall (Fig. 1B). In addition to increasing the respiratory surface area for gas exchange, the rugae in the epidermis of the body wall can also increase permeable surface area for sulfide to enter the body. Therefore, the location of SDO is important for counteracting sulfide via oxidation. Furthermore, the presence of rugae can also increase SDO contents by extending the area of the epidermis and is also helpful for uptake of the SDO substrate oxygen.

The SDO response to sulfide in the midgut and hindgut has been reported; the T-SDO SA in the midgut is kept stable by elevating the level of SDO protein while SDO in the hindgut displays a similar response pattern in both high and low concentrations of sulfide (*Zhang et al., 2016*). From the results of SDO response to sulfide in the gut, it is discovered that the specific enzyme activity of SDO protein is down-regulated by allosteric regulation and the T-SDO SA is maintained even elevated by the increasing the SDO contents by transcription regulation. However, the SDO response to sulfide in the body wall is

regulated by two different mechanisms to enhance the enzyme activity according to the sulfide concentration. At the high concentrations of sulfide (150 $\mu$M), the SDO activity (Fig. 4) increased with the increase in SDO protein contents (Fig. 3); furthermore, SDO protein expression patterns (Fig. 3) were consistent with mRNA level changes (Fig. 2). All of the results indicated that sulfide enhances SDO activity by promoting SDO mRNA transcription to elevate the SDO protein. However, in low concentrations of sulfide (50 $\mu$M), the mRNA levels and protein contents of SDO were both almost constant (Figs. 2 and 3), but the SDO activity was elevated significantly (Fig. 4), which indicated that sulfide might elevate SDO activity by allosteric regulation. GSH might play an important role in the allosteric regulation of SDO. Zhang et al. have proved that GSH can bind to the SDO protein (*Zhang et al., 2013*), and SDO activity in humans can be elevated by GSH (*Kabil & Banerjee, 2012*). From the above, when the concentration of sulfide is low, the sulfide might induce GSH binding to SDO to enhance the activity of sulfide oxidation. The response is quick, and the significant increase in SDO activity occurred at 6 h (Fig. 4). As the concentration of sulfide increases, GSH can be consumed to eliminate the reactive oxygen species induced by sulfide (*Pompella et al., 2003*) or by SQR catalyzing sulfide into persulfide (*Theissen & Martin, 2008*). Therefore, to enhance SDO activity, SDO protein contents were elevated after transcription was promoted. This response needs more time and is slow. It took approximately 48 h for the significant increase in SDO activity to occur.

## CONCLUSIONS

In summary, the epidermis of body wall in *U. unicinctus* is rugal, where the SDO protein located, which is important for counteracting sulfide via oxidation. When responding to sulfide, SDO activity can be enhanced for sulfide detoxification. The mRNA and protein expression analyses indicated that there are two regulation mechanisms for the increase of SDO activity: allosteric regulation at low sulfide concentrations and transcription regulation at high sulfide concentrations.

### Funding

This study was supported by the Natural Science Foundation of Shandong Province (Grant No. ZR2016CL17), Supporting Fund for Teachers' research of Jining Medical University (Grant No.JYFC2018KJ021), Doctoral Funding of Jining Medical University (Grant No. JY2015BS15), National Natural Science Foundation of China (Grant no. 31072191 and 31372506). The funders had no role in study design, data collection and analysis, decision to publish, or preparation of the manuscript.

### Grant Disclosures

The following grant information was disclosed by the authors:
Natural Science Foundation of Shandong Province: ZR2016CL17.
Teachers' research of Jining Medical University: JYFC2018KJ021.

Doctoral Funding of Jining Medical University: JY2015BS15.
National Natural Science Foundation of China: 31072191, 31372506.

## Competing Interests

The authors declare there are no competing interests.

## Author Contributions

- Litao Zhang conceived and designed the experiments, performed the experiments, analyzed the data, prepared figures and/or tables, authored or reviewed drafts of the paper, approved the final draft.
- Zhifeng Zhang conceived and designed the experiments, contributed reagents/materials/analysis tools.

## Field Study Permissions

The following information was supplied relating to field study approvals (i.e., approving body and any reference numbers):

The owner of the aquatic farm allowed our team to collect samples in his farms.

## Data Availability

The raw data is available as Supplementary Files.

## Supplemental Information

Supplemental information for this article can be found online at http://dx.doi.org/10.7717/peerj.6544#supplemental-information.

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
