# Peer review of "The response of sulfur dioxygenase to sulfide in the body wall of Urechis unincinctus"

_PeerJ, doi:10.7717/peerj.6544_

## Round 0.1 · original submission · Major Revisions

I have heard back from two reviewers, both of whom had many comments on your work. Reviewer 1 states the possibility of replicated work; and I have looked closely at the current and previous peer. While to me the two papers look similar, the raw data do look different. Still, I would appreciate you addressing this issue in detail, and failure to do so will likely result in rejection. Reviewer 2 also has many constructive comments; please consider these carefully. Finally, both reviewers find the English to be in need of work. Please provide me with the name of the proofreader (or service name) upon any resubmission.

Reviewer 1 ·

Basic reporting

The manuscript entitled, The response of sulfur dioxygenase to sulfide in the body wall of Urechis unicinctus by Zhang and Zhang explores the mechanisms use by Urechis to overcome the noxious properties of sulide. My comments are noted below.

Try and make some improvements to the text. The help of a native English speaking colleague could improve the use of the Scientific English used in the current manuscript. Similarly, some sections appear rather vague and would be strengthened by the inclusion of specific information. Indeed, in the background section the authors could add specific details like, ‘Sulfide is common toxin to organism. However, some organisms can tolerate certain concentrations of sulfide, such as the echiuran worm Urechis unicinctus, which inhabits in a U-shaped burrow in costal intertidal and subtidal mudflats, where sulfide can temporally accumulates’. In this instance this sentence could read, ‘In some sedimentary environments like costal intertidal and subtidal mudflats, sulphide levels can reach millimolar concentrations (2-5mM) and can be toxic to marine species. Interestingly, some organisms have evolved biochemical strategies to overcome and tolerate high sulphide conditions such as the echiuran worm, Urechis unicinctus. Other such changes are needed throughout the text.

A good use of references.

Experimental design

The experimental design is fine and is supported by previously published methods used by this group. See, Gene. 2016 Nov 30;593(2):334-41; PLoS One. 2013 Dec 2;8(12):e81885. doi: 10.1371/journal.pone.0081885. While the findings are interesting and go some way to explain a key mechanisms to overcome sulphide toxicity in this species, the results seem to replicate past work published in the aforementioned papers, Gene. 2016 Nov 30;593(2):334-41 and PLoS One. 2013 Dec 2;8(12):e81885. doi: 10.1371/journal.pone.0081885. Indeed, Figure 2 and 3 of this manuscript are similar to Figure 6-7 in the paper recently published in Gene. 2016 Nov 30;593(2):334-41. doi: 10.1016/j.gene.2016.07.045. Moreover, on close inspection the only differences are the units of expression for the activity data that is expressed as ng/µg protein as compared to µg/mg protein; essentially these are the same figures and this is would not represent new data or results. As such, can the authors describe how the current figures differs from past work and what additional novelty the current manuscript bring to this field base on their past work? Lastly, the graphs in Figure 2 A-C, Figure 3 A-C and Figure 4 A-C, clearly show differences between treatment groups but it may have been better to have combined each of these graphs into ONE graph for each Figure to aid comparison.

Validity of the findings

No comment

Additional comments

Please see text above.

Reviewer 2 ·

Basic reporting

I find that the paper, although readable, could benefit from additional translation efforts. There are numerous grammatical errors and typos, and some sentences could benefit from rewording/restructuring to improve clarity.

Additionally, I find that the introduction could improve with a reworked framework. The authors begin by stating the current relevance of hydrogen sulfide as a biological signaling molecule that is endogenously produced in organisms, but do not refer back to this concept at any point. Instead, I think it would best suit the paper to begin discussing the role of hydrogen sulfide as a physiochemical barrier to organisms via disruption of mitochondrial function, discuss the processes involved in sulfide detoxification, and then move into specifics of SDO and their study organism. Sulfide is an important challenge for many different environments which the authors have failed to mention here and this work could be relevant for those organisms as well, so I think these environments/systems should be introduced as well.

The hypotheses explicitly tested in this experiment are not outwardly clear, and the paper could benefit from explicit expectations. The question appears to be "how does SDO concentration, activity, and gene expression respond to sulfide concentrations in the body wall?". What do you expect? Why might that be? Zhang et al. 2016 found differences in expression of SDO upon the same sulfide exposures in the midgut and hindgut, does this influence your expectations? Additionally, lines 71-73 state that Menon and Arp indicate the importance of the body wall. In what ways? It might be best to include this background and state what your experiment will do to investigate this claim.

Lines 175-177 discuss the benefits of increased surface area of the body wall for oxygen uptake. Increased respiratory surface area can be beneficial for gas exchange but also increases permeable surface area for sulfide to enter the body. This may also be important for the localization of SDO activity.

Lines 180-183 are repetitive.

Experimental design

As stated above, I think the question and hypothesis could be more clearly defined. The research question is interesting, but I personally would like to see it related to the broader field of hydrogen sulfide biology and not just this system.

I think that the experimental design is reasonable and the methods are easily followed, but should be rewritten to improve clarity.

There is no mention of protein extraction, which should be included for both the ELISA and SDO activity steps.

Validity of the findings

Lines 142-143 state "Moreover, the negative control had no positive signals (Fig. 1C), which can prove the accuracy of the experiment." I would rewrite this. It definitely provides assurance that the results are accurate, but I am uncertain saying that it proves the accuracy is appropriate. That might just be semantics.

Each treatment was tested individually, but it might also be interesting to test for differences among treatments at each time point as well.

In the conclusion, it may be beneficial to reiterate what the question was and what your findings did to answer that question. Instead of listing out all the components of the body wall, state where the epidermis is located and why that's significant. Same goes for the SDO specifics, it would be good to see a brief explanation as to what these results mean.

Additional comments

I think this experiment and the results obtained are very interesting and provide great insights into the function of sulfide detoxification. It would be great to see the contents of this paper expanded to explain the benefits of this research to the greater hydrogen sulfide biology community.

---

## Round 0.2 · Minor Revisions

I have heard back from the two reviewers, who have found your paper much improved. There are some final small issues to address as per one of the reviewers, and thus my decision is minor revisions. I anticipate being able to receive your revision soon as the comments are very likely easy to deal with.

Reviewer 1 ·

Basic reporting

This version of the manuscript is much improved and has addressed the comments of each reviewer.

Experimental design

These are clearly described.

Validity of the findings

No specific comments

Additional comments

This version of the manuscript is much improved and has addressed the comments of each reviewer.

Reviewer 2 ·

Basic reporting

I think that the edits made to this paper successfully improved upon the comments made by the previous editors. The text readability has increased and there are few errors when it comes to the grammar of the paper. My comments are outlined as follows.

Lines 115-117: What is your isolation solution composed of? The solutions used in the RNA isolation and immunohistochemistry components of your methods are much more explicit, and for replication purposes it is necessary to include.

Lines 153-155: Revise wording. The emphasis here is that SDO is primarily expressed in the epithelial tissue, so the primary clause isn’t that necessary. If you decide to keep it, just reword it.

Line 190-192: The edited section here is composed of a complete and incomplete sentence. Just modify to ensure correct grammar.

227-233: The conclusion needs to be edited for clarity. The sentences are choppy. Tie in your findings to a broader audience, and I think this will be good.

Figure 1: there is no ER in the figure to represent epithelial tissue as indicated in the caption. Edit this to reflect, either by placing ER in the figure or editing the figure text to E to reflect epithelial tissue.

Experimental design

No comment

Validity of the findings

No comment

Additional comments

Overall, I think this is an excellent revision, and the intellectual merit of your findings are much more clear.

---

## Round 0.3 · Minor Revisions

I have gone over your edits; they are generally good from a scientific point of view. However, some small English edits are needed; I have included these in an attached MSWord file.

---

## Round 0.4 · accepted · Accept

Thank you for your patience in revisions. I am happy to accept this work and move it into production.